# The Effects of Crop Tree Management on the Fine Root Traits of *Pinus massoniana* in Sichuan Province, China

**Xiangjun Li** [1,2], **Yu Su** [3], **Haifeng Yin** [1,2], **Size Liu** [1,2], **Gang Chen** [1,2], **Chuan Fan** [1,2], **Maosong Feng** [1,2] **and Xianwei Li** [1,2,*]

[1] College of Forestry, Sichuan Agricultural University, Chengdu 611130, China; Lee_sicau@163.com (X.L.); yhfeng312@163.com (H.Y.); size_leo@126.com (S.L.); g.chen@sicau.edu.cn (G.C.); fanchuan@sicau.edu.cn (C.F.); fms444444@163.com (M.F.)

[2] Key Laboratory of National Forestry and Prairie Bureau on Forest Resources Conservation and Ecological Security in the Upper Reaches of Yangtze River (Sichuan Agricultrual University), Chengdu 611130, China

[3] Sichuan Academy of Forestry Sciences, Chengdu 610036, China; yusu110@163.com

[*] Correspondence: lxw@sicau.edu.cn; Tel.: +86-139-8160-5905

**Abstract:** *Pinus massoniana* is an important tree species for wind protection and timber forests in Southern China. In recent years, *P. massoniana* plantations have been developed on more than 11,300,000 $hm^2$ in southern China, but numerous problems have been observed, such as soil degradation, biodiversity reduction, and ecological functional decline. Crop tree management impacts on fine root development, which can be explained by the variations in the root orders. In this study, a 36-year-old *P. massoniana* plantation located in Huaying, Sichuan Province, was selected as the research field. In 2015, crop tree management was initiated, with a crop tree intensity of 150 trees per hectare. After 3 years of growth, fine roots of crop and noncrop trees were collected by the sector method with an angle of 15 degrees and a radius of 2 meters. We analyzed the morphological characteristics and biomass in different root orders, and explored their carbon and nitrogen contents. The results were as follows: (1) The specific root length (*SRL*), root length density (*RLD*), and surface root area (*SRA*) of the crop trees were larger than those of the noncrop trees; the *SRL* increased significantly from 0–0.5 m to 1–1.5 m from the stem. (2) The fine root biomass of the crop trees was significantly larger than that of the noncrop trees. The fine root biomass of the crop and the noncrop trees increased with the horizontal distance from the stem from 0–0.5 m to 1–1.5 m. The morphological indexes of the noncrop trees at the distances of 1–1.5 m and 1.5–2 m were significantly different, while those of the crop trees at those distances were not. (3) The fine root C content of the crop trees was significantly higher than that of the noncrop trees and varied significantly along a vertical distribution. The fine root N content of the crop trees was significantly higher than that of the noncrop trees, and the N content of topsoil was higher than that of deeper soil. In conclusion, our results indicated that crop tree management increased the production of a large-diameter wood of *P. massoniana*, which might be attributed to the improvement of soil permeability and nutrient stock, and thus, the enhancement of fine root quantity and water/nutrient absorption ability.

**Keywords:** biomass; crop trees; carbon and nitrogen content; fine roots; soil physical properties

## 1. Introduction

*Pinus massoniana*, an important shelterbelt and timber species in southern China, not only has a high economic value but also plays an important role in subtropical forest resources and ecological services in China. There has been a long-term adoption of the traditional pure forest management

model, which includes pest control, soil and water conservation, and tending thinning. Many problems have arisen, however, such as the decline in woodland fertility, species diversity, and ecological functioning [1]. To date, the reconstruction of low-efficiency plantations and vegetation restoration research have become important components of restoration ecology [2]. To achieve this, near-natural forestry measures such as crop tree management measures can be effective means to improve the productivity and production efficiency of *P. massoniana.*

The crop tree management system originated in Europe at the end of 19th century, and is a thinning method that aims to choose crop trees among forest stands and cut down trees that disturb the growth of crop trees to obtain large-diameter wood. Many countries have achieved a large number of cases of successful forest management [3,4]. Crop tree management suits the goals of foresters and can also be beneficial for producing better woods and achieving better economic returns. Moreover, crop tree management can take into account ecological environmental requirements and balance the multiple benefits of stands, which is closely connected with the National Forest Protection Projects in China and ecological civilization construction [5]. The most significant feature of crop tree management is that, unlike other types of forest management, it takes single crop trees as stand objects. By performing single-plant selection and management, we can not only support the ecological functions of the forest, but also maintain maximum accumulation and biomass [6].

The growth of the above-ground and underground parts of trees are closely related. The root system, as the link between the above-ground and below-ground parts, is an important part of the sustainable development of *P. massoniana.* Fine roots are the most active part of the root system and are the most important organs with which plants absorb nutrients and water [7]; fine roots are also important for carbon sequestration, so they are important carriers of material circulation and energy flow for stands [8]. Fine roots are especially sensitive to environmental changes, so most of the abiotic factors affecting woodlands significantly influence fine roots [9]. After being managed as crop trees, stands can free up space for crop trees to grow and affect the distribution of available soil resources. Fine roots respond rapidly in their morphology to changes in soil nutrient concentration. However, the morphological characteristics of fine roots are directly related to their functional characteristics [9,10]. Therefore, the morphological characteristics of fine roots are connected with terrestrial ecosystems and carbon and nitrogen cycles [11]. Research on the root system has been one of the most popular topics in ecosystem, ecology, and global change research [12,13].

This study was carried out in a 36-year-old *P. massoniana* plantation in a mountainous area in Huaying City, Sichuan Province. This study investigated the differences in fine root morphological characteristics and nitrogen and carbon distribution patterns between crop trees and noncrop trees, to reveal the growth rules of roots and provide theoretical support for the cultivation of *P. massoniana* large-diameter timber. At the same time, it provided a scientific basis for the reconstruction and sustainable development of *P. massoniana* plantations in the low hill districts of Sichuan Basin.

## 2. Materials and Methods

### 2.1. Study Area

The study area was located at the Tianchi Forest Farm, Huaying City, in the eastern Sichuan Basin (106°44′ 12′′ E, 30°16′33′′ N), at an altitude between 420 and 440 m. The area has a humid subtropical climate, the annual average temperature is 17.2 °C, and the annual average rainfall is 1087 mm. The soil in this experiment was typical yellow soil (Nitisol), the soil layer was thin, the soil fertility was low, and the soil was barren.

The *Pinus massoniana* plantation was established in 1982. Although necessary management measures have been undertaken in the stand, the overall management level has been low. Therefore, the ecological functioning of the plantation was in a degraded state, and the production function was very inefficient. The stand canopy density was 0.7, the average diameter at breast height was 18 cm, and the average tree height was 14 m. The understory plants were dense; the undergrowth

dominant shrubs were *Lindera glauca*, *Rubus chroosepalus*, and *Litsea cubeba*; and the dominant herbs were *Humata repens*, *Dicranopteris dichotoma*, and *Setaria plicata*. The crop tree operating density was 150 trees per hectare.

## 2.2. Crop Tree Management

Crop tree management was implemented in 2015. The basic principle for deciding on the density of crop trees is to ignore the crop tree distribution and choose as many trees as possible in the upper canopy (dominant wood and subdominant wood). The principle is as follows: The stands are averagely distributed away from forest roads and the edges of hauling roads or forest edges. The standard for selecting crop trees is that the trees must grow vigorously, trees with boughs must reach a height of more than 6 meters with a well-developed crown, and trees should be free of injuries caused by biotic or abiotic factors. After selecting the crop trees, the interfering and competing woody plants were removed from the sample plot, and all the trees except for the crop trees were designated as noncrop trees. After 3 years of management with these principles, three crop trees and three noncrop trees in each of three sample plots were randomly chosen. In total, 18 sample trees were selected. Basic situation of the crop and noncrop trees in Table 1.

**Table 1.** Growth status indicators of crop trees and noncrop trees.

|  | **Breast Diameter (cm)** | **Height (m)** | **Crown (m × m)** |
| --- | --- | --- | --- |
| Crop tree | 26.4 ± 0.6 | 18.5 ± 1.0 | 6 × 6 |
| Noncrop tree | 16.8 ± 0.8 | 15.2 ± 0.5 | 4 × 5 |

## 2.3. Sample Collection and Processing

For soil sampling around each sample tree, the sampling points were chosen approximately 0.5–1 m away from the base of the stem. Samples were collected at two depths (0–10 cm, 10–30 cm), each weighing approximately 1 kg, for the determination of chemical properties. Aluminum boxes were used to store the soil to determine the moisture content. Soil was taken with a ring cutter for the determination of soil bulk density. Soil samples were dried indoors and ground through a 2 mm sieve for the determination of chemical properties.

For collection of fine root samples for each sample tree, the base of the stem was taken as the center of a circle with a radius of 2 meters, and sector sampling was performed at an angle of 15 degrees. Samples were taken at 0–0.5 (I), 0.5–1 (II), 1–1.5 (III), and 1.5–2 m (IV) from the tree and at 0–10 cm and 10–30 cm soil depth (Figure 1). The fan-shaped slope dug out from under each sample tree was in the same direction as the slope. After taking the sample back to the laboratory, the soil was set on a 100-mesh soil sieve (aperture of 0.149 mm). The fine roots were carefully cleaned, avoiding breakage or loss [14].

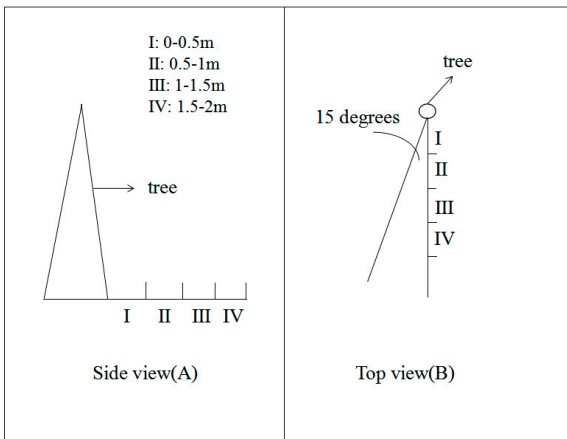

**Figure 1.** Schematic representation of fine root sampling.

### 2.4. Soil and Root Sample Analysis

Soil pH was measured in a 1:2.5 water and soil mixture whisked together for 10 min with a glass rod, before standing for 1 hour and then being measured with an electronic pH meter. The organic matter content was determined by the dichromate volumetric method/dilution heat method. Total soil nitrogen was determined by the semimicro–Kjeldahl method.

Systems of live fine roots were classified by root sequence location, named as described by Pregitzer [15]. A root with no branches at the far end of the root axis is called the first-order root. The first-order root grows from the second-order root, the second-order root grows from the third-order root, and so on up to the fifth-order root. Roots were carefully separated by root order with tweezers and placed in labeled glass dishes. The dried roots were not graded [16]. For determination of biomass, all grades were dried at 65 degrees to constant weight (48 h) and the dry weight of the fibrous roots was measured with a 1/10,000th microbalance.

### 2.5. Data Analysis and Processing

Specific root length (*SRL*), root length density (*RLD*) and surface ratio area (*SRA*) were calculated with the following formulas:

$$SRL = L/M \tag{1}$$

$$RLD = L/V \tag{2}$$

$$SRA = SA/M \tag{3}$$

$$B = M/d^2 \times 10000 \tag{4}$$

where *L* is the total length of each root system (m), *M* is the dry weight of the fine roots (g), *V* is the volume of clods (m$^3$), *SA* is the surface area of the fine roots (cm$^2$), *d* is the length of clods (cm), and *B* is the biomass of the fine roots (kg·hm$^2$).

The potassium dichromate oxidation–external heating method was used to determine the organic carbon in the fine roots, and the Kjeldahl method was used for total nitrogen determination.

Microsoft Excel 2016 and SPSS 25.0 were used for data processing and analysis, and Origin 8 was used to create the figures. Generalized Linear Models (GLMs) using a linear distribution and identity link functions were used to analyze the differences between crop and noncrop trees 3 years after treatment initiation, in terms of root order and sample distance, based on the morphological characteristics of fine roots. GLMs was also adopted to compare the effect of crop tree management and soil layer on soil physical and chemical properties.

## 3. Results

### 3.1. Fine Root Morphological Characteristics Following Three Years of Crop Tree Management Treatment

The fine root diameter of the crop trees did not differ from that of the noncrop trees (GLMs: Wald $\chi^2$ =1524.156, $p < 0.001$) (Figure 2). The distance to the stem did not affect the diameter of the fine roots (GLMs: Wald $\chi^2$ = 73.251, $p < 0.001$ for the distance). Moreover, the diameter of both the crop and noncrop trees decreased with the increasing root orders (GLMs: Wald $\chi^2$ = 3852.743, $p < 0.001$ for the crop orders). In other words, crop tree management had no effect on fine root average diameter.

The *SRL* of *P. massoiana* crop trees was found to be significantly increased compared to the noncrop trees after 3 years of crop tree management (GLMs: Wald $\chi^2$ = 1148.242, $p < 0.001$ for crop tree management) (Figure 3). Yet, the effect of crop tree management on *SRL* remarkably depended on distance (GLMs: Wald $\chi^2$ = 340.528, $p < 0.001$ for the interaction between crop tree management and distance). To be more specific, a crop tree at a distance of 1.0–1.5 m showed a similar *SRL* as one at 1.5–2.0 m, however, the *SRL* of a noncrop tree at the distance of 1.0–1.5 m was significantly higher than that at 1.5–2.0 m. Similarly, the effect of crop tree management on *SRL* remarkably depended on the root order (Wald $\chi^2$ = 126.232, $p < 0.001$ for the interaction between crop tree management and root

order), with a stronger increasing magnitude in the higher root orders than those in the lower root orders. Moreover, the *SRL* decreased with increasing root orders (GLMs: Wald $\chi^2$ = 4960.420, $p < 0.001$ for the crop orders).

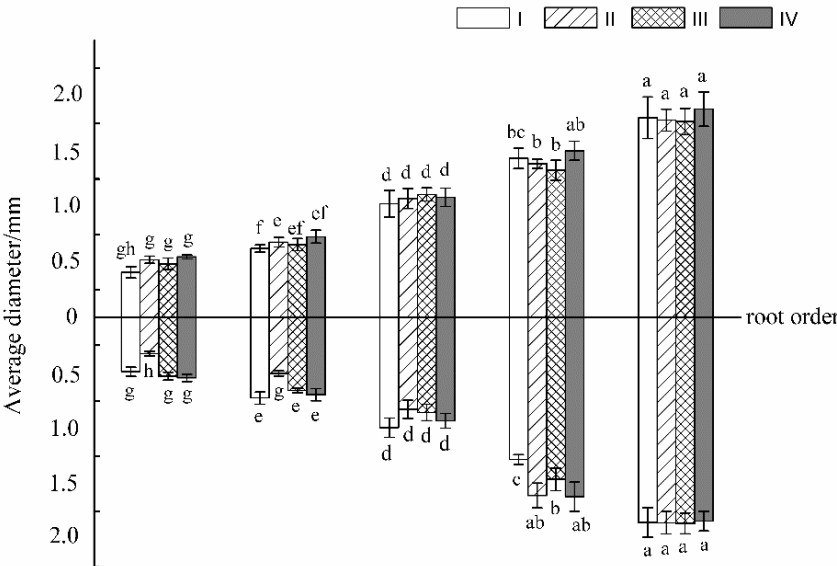

**Figure 2.** The effect of crop tree management on the root average diameter at different distances from the target *Pinus massoniana*. The X axis is the first to the fifth root order. The bars above the X axis represent the crop trees, and the bars below the X axis represent noncrop trees. I, II, III, and IV represent the distance of fine roots to the target tree at 0–0.5, 0.5–1, 1–1.5, and 1.5–2 m (this applies to Figures 1–5). Each bar represents the mean ± SD with three replicates. Different letters above or below each bar indicate significant differences among groups compared by Fisher's LSD test at $p < 0.05$.

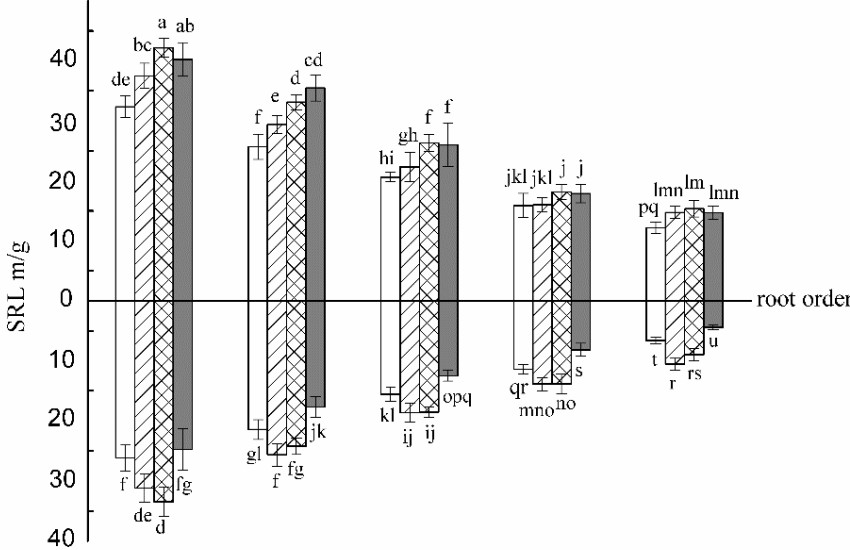

**Figure 3.** The effect of crop tree management on the *SRL* at different distances from the target *Pinus massoniana*. The X axis is the first to the fifth root order. The bars above the X axis represent the crop trees, and the bars below the X axis represent the noncrop trees. I, II, III, and IV represent the distance of fine roots to the target tree at 0–0.5, 0.5–1, 1–1.5, and 1.5–2 m. Each bar represents the mean ± SD with three replicates. Different letters above or below each bar indicate significant differences among groups compared by Fisher's LSD test at $p < 0.05$.

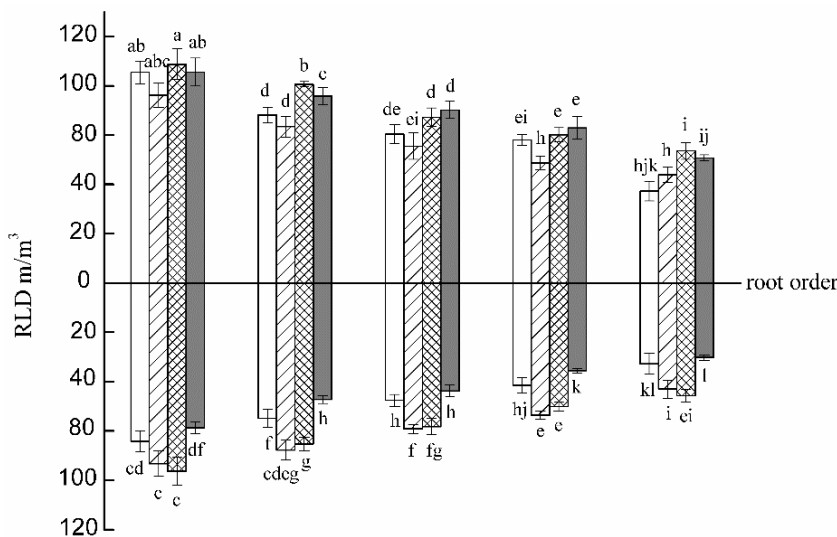

**Figure 4.** The effect of crop tree management on the *RLD* at different distances from the target *Pinus massoniana*. The X axis is the first to the fifth root order. The bars above the X axis represent the crop trees, and the bars below the X axis represent the noncrop trees. I, II, III, and IV represent the distance of fine roots to the target tree at 0–0.5, 0.5–1, 1–1.5, and 1.5–2 m. Each bar represents the mean ± SD with three replicates. Different letters above or below each bar indicate significant differences among groups compared by Fisher's LSD test at *p* < 0.05.

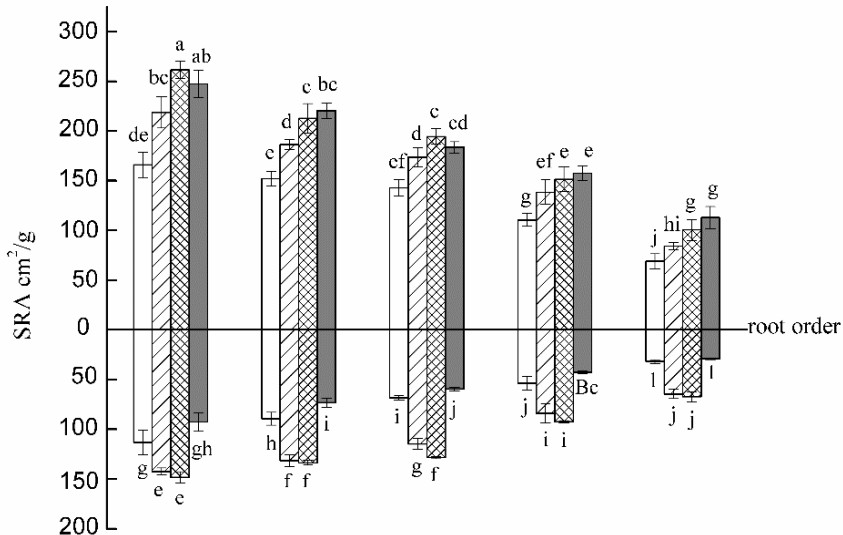

**Figure 5.** The effect of crop tree management on the *SRA* at different distances from the target *Pinus massoniana*. The X axis is the first to the fifth root order. The bars above the X axis represent the crop trees, and the bars below the X axis represent the noncrop trees. I, II, III, and IV represent the distance of fine roots to the target tree at 0–0.5, 0.5–1, 1–1.5, and 1.5–2 m. Each bar represents the mean ± SD with three replicates. Different letters above or below each bar indicate significant differences among groups compared by Fisher's LSD test at *p* < 0.05.

The *RLD* in *P. massoiana* crop trees was found to be significantly increased compared to the noncrop trees after crop tree management (GLMs: Wald $\chi^2$ = 1743.628, *p* < 0.001 for crop tree management) (Figure 4). The effect of crop tree management on *RLD* significantly depended on the root order as well (Wald $\chi^2$ = 482.593, *p* < 0.001 for the interaction between crop tree management and root order). To be more specific, a crop tree at a distance of 1.0–1.5 m showed a similar *RLD* as at 1.5–2.0 m, however, the *RLD* in noncrop trees at the distance of 1.0–1.5 m was significantly higher than at 1.5–2.0 m, which

is similar to the *SRL*. Moreover, the *SRL* decreased with increasing root orders (GLMs: Wald $\chi^2$ = 4134.227, $p < 0.001$ for the crop orders).

The *SRA* in *Pinus massoiana* crop trees was found to be significantly increased compared to noncrop trees (GLMs: Wald $\chi^2$ = 2437.026, $p < 0.001$ for crop tree management) (Figure 5). The fine root *SRA* increased with the increase in root order at each distance from the stem (GLMs: Wald $\chi^2$ = 667.624, $p < 0.001$ for the interaction between crop tree management and distance). The *SRA* of each root order showed significant differences at 0–0.5 m, 0.5–1 m and 1–1.5 m, and the trend presented was 0–0.5 m < 0.5–1 m < 1–1.5 m. As the distance increased from 1–1.5 m to 1.5–2 m, the *SRA* of the crop trees changed slightly, however, the *SRA* of noncrop trees decreased significantly. In summary, at farther distances, the fine root *SRA* increased first and then decreased, reaching its maximum at 1–1.5 m. The *SRL* decreased with increasing root orders (GLMs: Wald $\chi^2$ = 3058.475, $p < 0.001$ for the crop orders).

### 3.2. Fine Root Biomass Following Three Years of Crop Tree Management Treatment

The fine root biomass in *Pinus massoiana* crop trees was found to be significantly increased compared to noncrop trees (GLMs: Wald $\chi^2$ = 3125.726, $p < 0.001$) (Figure 6). The fine root biomass of both crop and noncrop trees increased with increasing root order, and there were significant differences among the root orders (GLMs: Wald $\chi^2$ = 4101.917, $p < 0.001$ for the root orders). As the distance to the stem increased from 0–0.5 m to 1–1.5 m, both the crop tree and noncrop tree fine root biomass increased (GLMs: Wald $\chi^2$ = 146.894, $p < 0.001$ for the interaction between crop tree management and distance). However, at 1.5–2 m, the change in crop tree fine root biomass was not obvious, but the noncrop tree fine root biomass decreased significantly. The fine root biomass increased with increasing root order (GLMs: Wald $\chi^2$ = 4867.334, $p < 0.001$ for the crop orders).

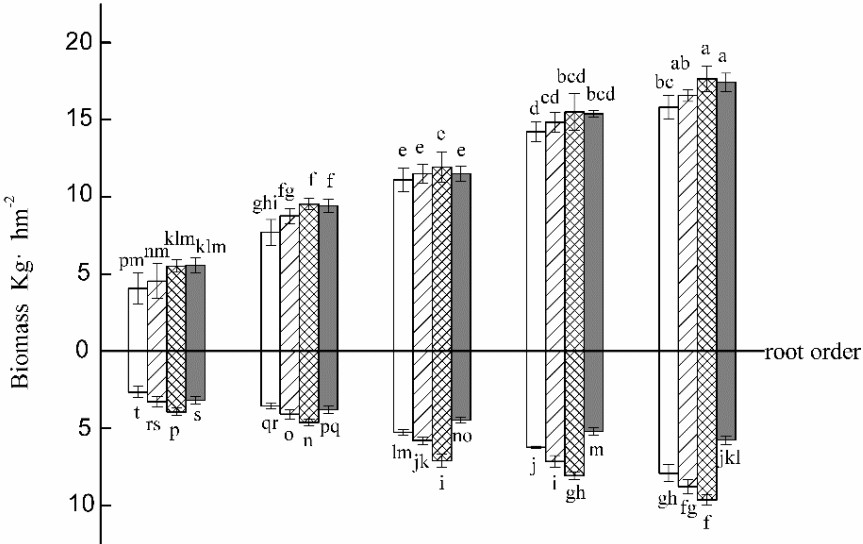

**Figure 6.** The effect of crop tree management on the root biomass at different distances from the target *Pinus massoniana*. The X axis is the first to the fifth root order. The bars above the X axis represent the crop trees and the bars below the X axis represent the noncrop trees. I, II, III, and IV represent the distance of fine roots to the target tree at 0–0.5, 0.5–1, 1–1.5, and 1.5–2 m. Each bar represents the mean ± SD with three replicates. Different letters above or below each bar indicate significant differences among groups compared by Fisher's LSD test at $p < 0.05$.

The soil depth significantly influenced the fine root biomass (Tables 2 and 3). The crop tree fine root biomass decreased with soil depth, and the biomass of first- through third-order roots decreased significantly ($p < 0.05$). Root biomass in the first- through third-order roots at 10–30 cm depth decreased by 3.31%, 4.4%, and 1.69%, respectively. The fourth- and fifth-order root biomass at 10–30 cm depth increased by 2.5% and 5.78%, respectively. For noncrop trees, the fourth- and fifth-order root biomass

increased by 3.41% and 2.95%, respectively, at 10–30 cm compared to that in surface soil, but the biomass of first- through third-order roots was reduced. Fine root biomass allocation to higher root orders increased with greater depth, and to lower root orders decreased with greater depth.

**Table 2.** Fine root biomass in each soil depth of crop trees.

| | 1-Order | | 2-Order | | 3-Order | | 4-Order | | 5-Order | | 1–5 Order |
|---|---|---|---|---|---|---|---|---|---|---|---|
| Soil | Biomass | Percent | Biomass | Percent | Biomass | Percent | Biomass | Percent | Biomass | Percent | Biomass |
| 0–10 cm | 6.87 ± 1.21b | 10.56 | 10.25 ± 2.36c | 15.8 | 13.18 ± 2.37c | 20.26 | 15.72 ± 3.15cd | 24.2 | 19.04 ± 2.12e | 29.3 | 65.06 ± 5.12a |
| 10–30 cm | 3.66 ± 0.85a | 7.25 | 5.74 ± 1.05b | 11.4 | 9.37 ± 1.95c | 18.57 | 13.49 ± 1.47d | 26.7 | 18.21 ± 1.51e | 36.08 | 50.47 ± 4.88a |

Values are mean ± SD. Different lowercase letters mean significant differences among each group at $p < 0.05$. The unit of biomass is kg·hm$^2$, and this also applies to Table 3.

**Table 3.** Fine root biomass in each soil depth of noncrop trees.

| Soil Depth | 1-Order | | 2-Order | | 3-Order | | 4-Order | | 5-Order | | 1–5 Order |
|---|---|---|---|---|---|---|---|---|---|---|---|
| | Biomass | Percent | Biomass | Percent | Biomass | Percent | Biomass | Percent | Biomass | Percent | Biomass |
| 0–10 cm | 4.78 ± 1.05b | 8.99 | 7.24 ± 2.26b | 13.62 | 11.51 ± 0.84bc | 21.65 | 13.16 ± 1.38cd | 24.76 | 16.47 ± 2.47d | 30.98 | 53.16 |
| 10–30 cm | 2.59 ± 0.27a | 6.01 | 5.36 ± 1.35b | 12.44 | 8.38 ± 1.73b | 19.45 | 12.14 ± 1.17c | 28.17 | 14.62 ± 2.08cd | 33.93 | 43.09 |

Values are mean ± SD. Different lowercase letters mean significant differences among each group at $p < 0.05$.

### 3.3. Soil and Fine Root Carbon and Nitrogen Concentrations

The crop tree fine root carbon concentration was higher than that of the noncrop trees (Tab 4). The carbon concentration in the first- through third-order roots was significantly higher than that in the fourth- and fifth-order roots in both crop trees and noncrop trees. Different soil depths resulted in different fine root carbon concentrations ($p < 0.05$), which were higher in the surface soil (0–10 cm) than in the 0–30 cm soil ($p < 0.05$). The crop tree fine root nitrogen concentration was higher than that of the noncrop trees (Table 4). At lower depths, the nitrogen concentration of the fine roots was significantly reduced ($p < 0.05$), and the surface soil nitrogen was much higher than that of the deep soil. The fine root nitrogen concentration of the noncrop trees at each soil depth did not obviously change ($p > 0.05$).

**Table 4.** Fine root carbon and nitrogen concentrations in *Pinus massoniana*.

| | Carbon (g/kg) | | | | Nitrogen (g/kg) | | | |
|---|---|---|---|---|---|---|---|---|
| | Crop Tree | | Noncrop Tree | | Crop Tree | | Noncrop Tree | |
| Soil Depth | Lower Orders | Higher Orders | Lower Orders | Higher Orders | Lower Orders | Higher Orders | Lower Orders | Higher Orders |
| 0–10 cm | 675.33 ± 6.37a | 752.67 ± 11.36c | 631.67 ± 5.48d | 713.46 ± 12.1f | 18.57 ± 1.14a | 15.26 ± 0.94b | 13.44 ± 1.32b | 11.28 ± 0.73d |
| 10–30 cm | 624.85 ± 4.35 ± b | 673.33 ± 8.41a | 542.18 ± 6.72e | 619.25 ± 8.58b | 13.34 ± 1.51b | 10.53 ± 0.65c | 12.39 ± 1.18b | 8.12 ± 0.46e |

Values are mean ± SD. Different lowercase letters mean significant differences among each group at $p < 0.05$. Lower orders are the first to third orders, higher orders are the fourth and fifth orders.

The soil pH of the crop trees and noncrop trees was approximately 5 in each stand (Table 5), which is a typical soil acidity. In terms of soil ρb, that of the crop trees was higher than that of the noncrop trees, and that of the surface soil was higher than that in the deep soil. Soil porosity changed in the same way as soil ρb. This means that crop tree soil is more breathable and easier for water to permeate than noncrop tree soil. Changes in the soil nutrient content indicated that the soil of the crop trees had a higher organic carbon concentration than that of the noncrop trees, and there was a significant

difference at 0–10 cm ($p < 0.05$). The soil nitrogen concentration of the crop trees was higher than that of the noncrop trees ($p > 0.05$).

**Table 5.** Soil physical and chemical properties.

| | Depth | pH | ρb | Porosity | Water Content | Organic Carbon | Nitrogen |
|---|---|---|---|---|---|---|---|
| | | (H2O) | g·cm$^3$ | % | % | g·kg$^{-1}$ | g·kg$^{-1}$ |
| crop trees | 0–10 cm | 4.661 | 1.26 | 41.56 ± 1.21a | 13.4 ± 1.36a | 36.55 ± 2.15a | 1.87 ± 0.24a |
| | 10–30 cm | 4.657 | 1.04 | 37.22 ± 0.69b | 8.78 ± 1.52b | 25.1 ± 2.67b | 1.72 ± 0.08a |
| noncrop trees | 0–10 cm | 5.057 | 1.12 | 36.14 ± 2.73b | 11.35 ± 0.54c | 32.25 ± 1.89c | 1.65 ± 0.02a |
| | 10–30 cm | 4.983 | 1.06 | 32.57 ± 1.65c | 6.47 ± 1.71b | 24.44 ± 1.05b | 1.55 ± 0.06b |

Values are mean ± SD. Different lowercase letters mean significant differences among each group at $p < 0.05$.

There was a significant negative correlation between fine root *SRA* and ρb and the soil nitrogen concentration ($p < 0.05$) (Table 6). There was also an extremely significant positive correlation between fine root nitrogen and soil porosity ($p < 0.01$), and a significant negative correlation between fine root nitrogen and ρb ($p < 0.05$).

**Table 6.** Correlation coefficients between fine root and soil properties.

| | pH | ρb | Soil Porosity | Water Content | Organic C | N |
|---|---|---|---|---|---|---|
| *SRL* | 0.06 | –0.17 | 0.25 | –0.23 | –0.25 | –0.33 |
| *RLD* | –0.14 | 0.02 | –0.08 | –0.51 | 0.09 | –0.23 |
| *SRA* | 0.25 | –0.42* | 0.11 | –0.21 | –0.18 | –0.17* |
| Biomass | –0.35 | 0.33** | –0.38* | –0.05 | 0.46 | 0.56 |
| Fine root C | 0.13 | –0.24 | 0.01 | 0.64** | –0.14 | –0.02 |
| Fine root N | 0.07 | –0.26* | 0.17** | –0.07 | –0.29 | 0.37 |

Asterisks indicate significant results.* $p < 0.05$, ** $p < 0.01$.

## 4. Discussion

### 4.1. Crop Tree and Noncrop Tree Fine Root Morphology

The fine root diameter difference among different root orders within the same horizontal distance was significant for both crop and noncrop trees after crop tree management. As the distance from the base of the stem increases, the root order diameter of the crop tree does not change significantly. The diameter of the first- through third-order fine roots of noncrop trees is obviously reduced when the horizontal distance increases from 0–0.5 m to 0.5–1 m. This may be due to the feedback mechanism of noncrop trees to increase the absorption competitiveness of fine roots in the ground. *SRL* is the ratio of root length to biomass. A longer root length means higher biomass input efficiency of this root order, and *SRL* is inextricably linked to fine root absorption functioning [9]. In this experiment, the *SRL* of the crop trees and the noncrop trees increased gradually with increasing horizontal distance from 0–0.5 m, to 0.5–1 m, to 1–1.5 m. There was no significant difference between 1–1.5 m and 1.5–2 m in crop trees, but the *SRL* of noncrop trees was significantly reduced. The difference between crop and noncrop trees is the embodiment of the different resource allocations of the crop trees after crop tree management. Root length density (*RLD*) is the total length of the fine roots per unit volume, and a higher *RLD* represents higher absorption efficiency of water and nutrients [17]. After crop tree management is implemented, the *RLD* of the fine roots of the crop trees is increased, and the increased magnitude of first- through third-order roots of the crop trees is greater than that of the fourth- and fifth-order roots, which is compatible with the physiological functions of the lower roots. In the horizontal direction, the change of *RLD*, as well as *SRL*, is significantly reduced in noncrop trees from 1.5–2 m away from the base of the stem. The specific surface area (*SRA*) characterizes the absorption efficiency of nutrients in the fine root biomass, and a larger *SRA* of fine roots means higher competitiveness for underground

nutrients [18]. Crop tree management, as a special thinning style [19], releases the growth space of the crop tree *P. massoniana* in the plantation, changing the soil nutrient status and biodiversity of the forest, and promoting the fine roots of the crop tree to absorb water. The effective competition for nutrients, in turn, changes the growth strategy of the fine roots of the crop tree to increase the root length, root length density, and specific surface area to achieve higher utilization efficiency.

### 4.2. Fine Root Biomass

The fine root biomass is the quantity of fine roots per unit area and is subject to a variety of conditions, including tree species, community structure, climate, and soil physical and chemical properties [20]. Our study has shown that fine root biomass increased significantly with increasing root order, indicating that the fine root order of *P. massoniana* is the key factor affecting fine root biomass. From a physiological point of view, the fine roots of *P. massoniana* significantly increase the distribution of biomass of higher roots to absorb water more quickly from the soil, which is related to the adaptability of the tree species to the environment. The distribution of the fine root biomass of crop trees in the horizontal direction is consistent with the distribution of fine root diameter, which is 0–0.5 m < 0.5–1 m < 1–1.5 m, while the fine root biomass at 1–1.5 m and 1.5–2 m is not consistent with the distribution of the fine root diameter. The growth of the whole root system can be inferred from the biomass distribution of the fine roots [21]. The lateral growth of fine roots in the horizontal direction broadens the growth range of the tree, and the increased biomass results from the increase in the upper part of the diameter of *P. massoniana*. The tendency of root biomass to be stable at 1.5–2 m indicates that the fine roots of the crop tree still maintain high competitiveness in the underground space, which contributes to the cultivation of large-diameter timber in *P. massoniana* plantations. The fine root biomass of noncrop trees was significantly lower than that of the crop trees. The variation in fine root biomass at 0–1.5 m was similar to that of the crop trees, but the 1.5–2 m fine root biomass was significantly lower than that at 1–1.5 m. This may be due to insufficient competitiveness and compression of growth space, as the tree is unable to maintain strong growth at the distal root. In the vertical direction, the availability of soil resources leads to structural and functional changes in the fine roots [15]. The basic response to changes is to adjust fine root biomass, which is also the fine root adaptation to environmental spatial changes [22]. In this experiment, with the deepening of the soil layer, the fine root biomass of the crop tree decreased from 65.06 kg·hm$^{-2}$ to 50.47 kg·hm$^{-2}$, and that of the noncrop tree decreased from 51.16 kg·hm$^{-2}$ to 43.09 kg·hm$^{-2}$. This reflects the vertical distribution characteristics of the fine roots of the *P. massoniana* plantation in the crop tree management model. The topsoil is enriched with fine roots, and their biomass is higher than that at deeper layers. *P. massoniana* is an early-succession tree species, and the characteristics of fine root adaptation are characterized by rapid occupation of soil space and expansion of the growth range. The biomass per unit volume of clods is lower, and the distribution in the vertical direction is more uniform [23]. The results of this experiment are different from those found in the related research. This may be due to differences in site conditions and in soil physical and chemical properties.

### 4.3. Fine Root Carbon and Nitrogen Content

The first- through third-order (lower roots) fine roots appear as a complete functional module with significant differences in C, N content, turnover rate, respiration rate, and more from those of the fourth- and fifth-order roots (higher roots) [10]. The carbon content of the fine roots of *P. massoniana* under crop tree management showed obvious vertical differences. The content of fine root carbon in topsoil at 0–10 cm was significantly higher than that at 10–30 cm, which indicated that the fine root carbon content of the *P. massoniana* plantation was enriched in surface soil. This phenomenon is consistent with the research results of Meier [24]. The fine roots of *Acacia crassicarpa* and *Casuarina equisetifolia* in the coastal areas of southeastern Yunnan are mainly distributed in the soil surface, and there is a significant positive correlation between organic carbon and fine root biomass in each soil layer [25]. The fine root organic carbon content of the crop trees was higher than that of the noncrop trees, which

indicates that the crop trees have a higher absorption capacity than the noncrop trees in the lower roots, and strong transport capacity and stress resistance in the upper roots [9,26]. However, after the disturbance of tree removal, the illumination increases, the surface temperature of the crop canopy rises, and the humidity increases, which accelerates the decomposition of litter under the crop forest, leading to an increase in soil carbon storage, which is characterized by fine root carbon content [27]. In this experiment, the fine root N content of the first- through third-order roots in the *P. massoniana* plantation was significantly higher than that of the fourth- and fifth-order fine roots. The fine root N content corresponds to the respiration rate, and the higher N content means a higher respiration rate and stronger activity [28]. Studies on the fine roots of *Cupressus funebris* also found that the N content in the fine roots decreased with the increase in root order [29]. This N distribution model in fine roots is more conducive to fine root turnover, which is reflected in the production and death of fine roots. The fine root N content of the crop trees was higher than that of the noncrop trees in each soil layer. After this management, the crop trees are cultivated as single plants under conditions created by human disturbance, and the growth condition of the forest is also reflected in this. With the high vitality of fine roots, higher N content shortens the renewal cycle of fine roots and maintains high absorption, transport, and storage functions in the long-term development of forest trees [30,31].

## 5. Conclusions

After we investigated the root system of *P. massoniana* under the management of the crop trees, we found that the fine root biomass of crop trees increased significantly, and crop tree fine roots had higher productivity than those of non-crop trees at the far end of the tree stem, which means this management had a profound effect on increasing forest carbon stocks and nitrogen cycle promotion. This study revealed that crop tree management is positive and effective for large diameter timber cultivation of *P. massoniana* through the improvement of underground fine root quantity and water/nutrient absorption. This study emphasized the study of individual trees. In the future, it will be useful to explore the growth of forests after crop tree management.

**Author Contributions:** Conceptualization, X.L. and Y.S.; methodology, H.Y.; software, G.C.; validation, X.L., Y.S. and S.L.; formal analysis, M.F.; investigation, X.L.; resources, X.L.; data curation, H.Y.; writing—original draft preparation, X.L.; writing—review and editing, G.C. and C.F.; visualization, G.C.; supervision, X.L.; project administration, X.L.; funding acquisition, X.L. All authors have read and agreed to the published version of the manuscript.

**Funding:** This study was funded by a Pillar Project of the "13th" Five-Year Plan for **C**hina (grant number 2017YFD060030205), and German Government Loans for Sicuan Forestry Sustainable Management (grant number G1403083).

**Acknowledgments:** This study was supported by the Project as above. We also thank all professors who provided helpful guidance in this research.

**Conflicts of Interest:** The authors declare no conflict of interest.

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
