# Peer review of "The Effects of Crop Tree Management on the Fine Root Traits of Pinus massoniana in Sichuan Province, China"

_forests, doi:10.3390/f11030351_

Round 1
Reviewer 1 Report
In the aim of the study, you emphasize that the research was done on root order. However, the description does not indicate how the obtained results are related to the root orders.
Even if the research did not focus on the mycorrhiza determination, it is worth mentioning that it was not the first-order roots that were not morphotyped.
Table 6: Please indicate in the description was matched by “*”
Author Response
Thank you very much for your suggestion.I've revised my manuscript(Table 6).
(In the aim of the study, you emphasize that the research was done on root order. However, the description does not indicate how the obtained results are related to the root orders).I have described the result consequence about SRL,RLD and SRA with root orders,but in this study,What I attention more is the difference between the crop tree and the noncrop tree.
(Even if the research did not focus on the mycorrhiza determination, it is worth mentioning that it was not the first-order roots that were not morphotyped).
Yes,I've deleted it,this statement is unscientific.
Thanks again and best wishes to you and your family.

Reviewer 2 Report
See attached documents on general comments and a marked PDF.
General Comments (v1)
The authors improved the manuscript but there continue to be major problems that if not addressed,
make it unsuitable for publication.
The fundamental problem is that the conclusions stated about effects of crop tree management cannot
be made due to limitations in the methodology. This was pointed out in the review of the original
manuscript but the authors did not provide new information on methods enabling these conclusions
and do not appear to recognize the problem.
While the research demonstrates differences between crop trees and non-crop trees three years after
the initiation of crop tree management, there is no recognition that these differences cannot, with any
confidence, be attributed to crop tree management. The differences observed may be due to initial
differences in tree size (not controlled experimentally or accounted for in the analysis), crop tree
management, or the combination of the two. The large differences in sample tree DBH, HT, and crown
size three years after crop tree identification and initiation of crop tree management (Table 2) suggest
that initial size differences (crop trees on average larger in size than non-crop trees) between crop tree
and non-crop trees sampled were considerable. Initial tree size likely had a major impact on all the
reported tree attributes three years after crop tree identification and crop tree management initiation.
There are conclusions that can be made based on the methods and results. It can be concluded that
attributes between crop and not crop trees differed and that these differences may have resulted from
initial size differences, crop tree management or their combination. It can be concluded that crop tree
attributes promote ecosystem health and function based on other literature.
The writing is improved from the original manuscript but still requires more careful review and revision.
See suggested edits through line 149 on marked pdf for examples of revisions needed.
Note that I did not review lines 150 through 363.

Author Response
Thanks for your suggestions.I've revised my manuscript according to the PDF you noted.I can understand what you mean,the differences between crop and noncrop tree were not because of the crop tree management,it was maybe the difference about trees at the start of the management.I've considered this point at the start of this study.In the study area,the average diameter at breast height was 18 cm, and the average tree height was 14 m(Materials and Methods),all trees in the plantation planted in the same year,so there were not significant differences among the trees.After crop tree management for three years,the crop trees were much higher and thicker than noncrop tree,so we can believe that this management cause the differences,otherwise,the methods of sample collection were random and repetitive,it also excludes the effect of the trees themselves on the experimental results.This is my explanation of your question.I've asked MDPI for English Editing and finished.Please check it again.Thanks for all your help and best wishes to you and your family.

This manuscript is a resubmission of an earlier submission. The following is a list of the peer review reports and author responses from that submission.
Round 1
Reviewer 1 Report
Although this paper seems to show interesting data, I cannot recommend that it will be accepted for publication in Forests in this present form because of unclear objectives and insufficient information on study design. Please see my comments listed below.
1) There are many awkward sentences. Please be checked by native speaker or professional language editor.
2) Title
I suppose that this would be “The effects of crop tree management on fine root traits of Pinus massoniana in Sichuan Province, China”.
3) Abstract
L11, The crop tree management as an effective…: I cannot understand what this sentence means very well.
4) Introduction
Please show objectives and hypotheses of this study. I do not understand well why you examined fine root traits (biomass and morphology) at different distances from stems of sample trees. What did you expect by examining the fine root traits in such a way in investigating effects of tree crop management?
5) Materials and methods:
Please explain design of this study in more detail. What was the size of each study plot (area) where three crop and non-crop trees were present? What was stand density of the sampling plots? What were conditions of understory vegetation? How many trees around a crop tree were removed (or what area around the tree was cleared) in the crop tree management? What were the distances between sample trees or what were the distances from the sampling area (within 2 m from a trunk) of a sample tree to neighbor trees (or neighbor sample trees)? These information on spacing are important to consider whether the fine root data could be affected by neighbor trees if fine root sampling was done with soil coring or a similar method. Or if you identified roots of sample trees, please show how you did it. Although you showed sampling locations of fine roots in Fig. 1, I do not understand well what the “an angle of 15-20 degrees (L104)” means. What is the line 15 degrees away from the sampling line (I to IV) in Top view (B) of Fig. 1? Please clarify how you sampled fine roots. If it was soil coring or soil block sampling, please show the size of the soil samples. As for soil analyses please show how you measured soil porosity and ρb (bulk density?).
6) As a whole, it is unclear for me what you focused on in this study as I commented above for Introduction. Please show objective and hypothesis of this study more clearly and discuss the results related to them. I need to know sampling design in more detail to see whether the results of this study were reasonably discussed. However, if I add a few comments, I would like to suggest that you will consider root tissue density to explain variation of SRL and SRA because there were no significant changes in fine root diameter between the treatments or across the varied distances from the trunks (see e.g. Ostonen et al. 2007 Plant Biosystems 141: 426-442).
7) Other comments
L19, SRA: This would be “Specific root area”.
L69 and others, 36a: 36-year-old
L80: What was the “some necessary operation measures”?
L113, 1:2.5 water and soil mixture: 10 g soil in 25 mL water?
L120, oldest roots without branches: How did you know these were “oldest”?
L123, one ten thousandth of a gram: 10-4 g?
L126: SRL, RLD and SRA should be written in full spelling here.
L134, The potassium…: This sentence should be moved to previous section (2.4).
Fig. 2: Please indicate root order (1st to 5th) in the figure.
Figs. 3-6: Please indicate root order (1st to 5th) and sampling locations (I to IV) in the figures.
Tables 2-4: Please show units of the data.
L284, changed the soil nutrient…: no evidence in this study.
L295, The distribution of…: I do not understand what this sentence means.
L312, This reflects…: It was unclear for me how the crop tree management is related to vertical distribution patterns of fine root biomass, which appeared to have no big differences between the crop and non-crop trees.
L330, The fine root organic carbon…: I think that there was no evidence for relationships between the fine root organic carbon content and fine root physiological activities as written here.
Reviewer 2 Report
The effects of fine root after crop tree management on Pinus massoniana in Sichuan Province, China
General comments
The topic raised in the manuscript is extremely interesting and important. The authors introduce the purpose of the study very well, and ended with clearly stated hypotheses. However, I suggest to underline how the chosen traits impact nutrient acquisition. In the abstract, the authors should indicated, which of the traits changes within root order, which is very important. Some more specific comments I attached below.
The manuscript in its present form requires revision before it can be acceptable for publication.
Specific comments
Abstract
Line 9-11: do you want to say that these problems were created by P. massioniana plantations?
In the aim of the study, you emphasize that the research was done on root order. However, the description does not indicate how the obtained results are related to the root orders.
There is only a statement related to top and deeper soil, but you do not refer to soil physical properties.
Introduction
The introduction emphasizes the importance of morphological traits in nutrient acquisition. There is no indication, however, how changing these characteristics affects acquisition. In addition, specific root features were used in the study. Why were these chosen?
Materials and Methods
Line 105: Add “m” after “1.5-2(â…£)”.
Line 116: You should either add citation or describe the method in detail.
Line 118: Generally agree with the description. However, the most distal root order may have branching due to mycorrhizae, frequently dichotomize branches. You should consider it in your description, unless you had no mycorrhiza, which is rather rare in situ.
Line 120: “why do you describe the first root order as the oldest one?
Line 134-135: Again, you should add some details of the methods use in the study.
Line 140: You should here also recall the test used for comparison of the mean for all of the root traits.
Discussion
Line 262-263: This sentence is in contradiction to what was described in Results.
Line 265-266: It is two general, and true only for the II distance class
Line 272: reduce in reference to what? At all or in particular distance class.
Conclusion
You could indicate some future direction of the study to build better scientific foundation the reconstruction and sustainable development of P. massoniana plantation in the low hills.
Figure 1
I do not understand the scheme. Please explain what do you mean by I, II, III, and IV. This Figure is not self-explanatory.
Tables
Table 2, 3, and 4. You should add units in which biomass was measured and the test used for comparison the means.
Table 5: which test was used for comparison the means?
Table 6: Did you use Pearson coefficient? It is “r” I believe. What was marked by “*”
Reviewer 3 Report
General comments
The manuscript provides a valuable contribution in describing Pinus massoniana tree size, fine root, and soil attributes for crop and non-crop trees three years after initiation of crop tree management competition removal around crop trees.
However, the principal conclusion, “This study reveals that crop tree management is positive and effective for large diameter timber cultivation of P. massoniana through the improvement of underground fine root quantity and water/nutrient absorption” cannot be made due to limitations in the research approach and experimental design as described in the manuscript. See line comments for more detail.
The manuscript needs significant revision to either more fully describe the methods used that would support the current conclusions or change the narrative in portions of the introduction, discussion and conclusion sections to emphasis the differences observed between crop trees and non-crop trees and their associated fine roots and soil environment three years after initiation of crop tree management and remove statements that cannot be supported regarding effects of crop tree management.
Revisions are needed in all sections of the report.
The Introduction will benefit from better description of terms and an explanation of why a crop tree and non-crop tree comparison is of value.
The Methods section needs much more detail on the study area, stand attributes, plot establishment and attributes, and sample tree selection.
The Results section would improve with correct placement of figures and tables and reformatting of tables.
The writing requires additional editing to be more direct and concise.
Line
Abstract
2 Title should be rewritten. The title is not clear and inconsistent with the methods described. The methods described resulted in quantifying tree size, fine root, and soil properties of crop and non-crop trees three years after crop tree management. The methods do not allow any attribution of “effects” of fine roots on Pinus massoniana or effects of crop tree management on fine roots.
9 “…have been greatly developed..” is vague. Change to be more descriptive such as (…has been planted on more than xxx hectares>>>”
The sentence, “ The crop tree management….” is not clear. Rewrite. For example, “Crop tree management impacts fine root development as well as aboveground tree growth.” Change order. First write full word description then abbreviation in parenthesis e.g. specific root length (SRL) at first mention. Generic comment to apply elsewhere.
22 Delete “In the horizontal direction,..”
28 The conclusion stated in the last sentence cannot be made given the description of the methods used. Differences between crop and non-crop trees in tree size, fine roots and soil properties may result from differences in tree size, fine root distribution and tree specific soil quality that existed prior to the crop tree management treatment. Even if the actual methods did allow attribution of three year post treatment differences to crop tree management, the premise of the conclusion is questionable and needs greater context. Crop trees have more aboveground and belowground growing space. This provides more resources per tree even if soil quality was not improved. This is as important or more important than environmental quality (soil permeability and nutrient stock) mentioned as the primary driver enhancing fine root quantity and absorption ability.
Introduction
37 “Traditional pure forest management model” meaning will not be clear to many readers. Be more descriptive.
39 Be more descriptive. What is meant by “low-efficiency plantations”?
41 What does “this” refer to. Rewrite. What is near-natural forestry and how is it a crop tree management measure? Unclear. Rewrite.
45 The end of the sentence starting with “….and many….” Is not clear. Rewrite.
53 What does “maximum woodland production” mean? Rewrite to state what this means.
55 The root system is an underground part. How can it link itself to aboveground parts? Rewrite.
59 This basically is a repeat the point made in line 57. Reword or delete.
60-61 The statement the “all abiotic factors affecting woodlands always significantly influence fine roots” is an unprovable absolute. Restate.
63-64 Be more direct and clear in writing e.g. change to “Fine roots respond rapidly in their morphology to changes in soil nutrient concentration”.
69-75 Rewrite to be more direct and clear. It is not clear from the introduction why it is important to examine differences in fine root morphology and function between crop and non-crop trees.
Materials and Methods
78-83 This section is incomplete and the writing is not clear. What does the “a” in 36a refer to? Describe important site and stand characteristics (latitude, longitude, climate soil taxonomy, age, basal area, trees per Ha), past operational and management efforts. Most readers have no idea what is considered a “typical site” for crop tree management of this species in a plantation in this region. State if there is any vegetation other than P. massoniana?
85-95 This section is not clear and incomplete. Rewrite. It is implied that some trees do not have branches (boughs). That seems in error. How were the locations of the three plots selected within the stand? How large where the plots? Where stand and tree attributes assessed in each stand prior to or immediately following competition removal around crop-trees? How was the random selection of the three crop trees and three non-crop trees done? Describe details (area, how, when) of the competition removal around crop trees.
NOTE: The methods described do not allow any attribution that differences in root and soil properties between crop and non-crop trees three years post treatment was the “effect” or “result” of crop tree management. To make this assertion,, there would be plots of a control treatment (plots and trees of crop tree quality not receiving competition control) and effects of crop tree management evaluated by comparing attributes of “crop trees” with and without competition removal. Alternately it would be possible to make this assertion if there were measurements (DBH, Ht, Crown) of the sample trees (essential) and soils (preferred but not essential) around sample trees prior to the crop tree management treatment and you controlled or took into account differences in pretreatment tree size and site quality prior to treatment.
95 Table 1 is a result not a method. Move to results and provide a paragraph on the results. Smaller size of non-crop trees as compared to non-crop trees is a big result that influences all the belowground results. Impacts of tree size differences between crop and non-crop trees at time of competition removal around crop trees is important to provide context to all the results reported,
97 Use a standard format of topic sentence starting the paragraph rather than Soil Samples: each..
98 These are soil depths not tiers.
103 Use standard format of topic sentence starting the paragraph rather than Fine root samples: each..
105 State that the distances are meters.
113 Use standard sentence format throughout rather. As an example do not use “Soil pH:a…”
117 Change “..roots are classified..” to “..roots were classified..”
126 Spell out full name before using abbreviation.
131-132 Clarify “volume of clods” term. There is no mention of clods in the sampling section. What is a clod?
138 Given the methods described, there is not ability to analyze for effect of tree class (crop tree or non-crop tree) but not for effect of crop tree management. Make this change throughout manuscript.
139-140 State interactions evaluated.
141 Change form “Analysis and Results” to “Results”
144-145 Given the methods described, no inference can be made that differences in root properties between crop and non-crop trees if the effect of crop tree management. Change title and specify results are for three years after crop tree management treatment. Comment applies to Figures 1 to 5.
147-148 Change to “…the distance (0-0.5m, ………), respectively of fine roots from the target tree. Each..” This applies to Figures 1, 2, 3, 4, and 5 titles.
149 This and all similarly formatted figures should have the legend indicating the distance from tree classes. Consider putting Figures 1-5 in one compound figure.
172 Delete”for crop tree management”.
172 Be consistent in stating distance from tree as described in Methods. Do not state distance to the tree. Apply throughout.
Use “tree” or “stem”. Avoid using “trunk”. Delete the sentence starting with “In other words..” It repeats line 171.
214 and 216 What are the units for biomass? Reformat to have means and SD on same line. Consolidate into one table.
219 This paragraph makes no mention of the greater fine root biomass for crop trees as compare with non-crop trees across root orders and distances from stem. Isn’t this a principal result?
221 Change “crop orders” to “root orders”
224-226 The statement starting “The fine root biomass decreased….” Is opposite what Figure 6 demonstrates and the first sentence of the paragraph.
229-232 The text misrepresent the fine root biomass allocation by root order with soil depth. State that fine root biomass allocation to higher root orders increased with greater depth and to lower root orders decreased with greater depth.
245 Improve table formatting e.g. all attributes presented refer to the soil. No need to have the word soil in column headings.
235 and 236 Units for carbon and nitrogen concentration? Reformat column headings. Consolidate into one table.
254 Change title to “Correlation coefficients between fine root and soil properties”
Discussion
272 -287 It is impossible to attribute the differences observed in fine roots and soil properties to the effect of crop tree management given the description of the study in Methods. It would be possible to make this assertion if there were measurement for the sample trees and soil around sample trees prior to the crop tree management treatment and you controlled or took into account differences in pretreatment tree size and site quality prior to treatment. It cannot be inferred from the results presented (only three years post treatment) that difference resulted from crop tree management. The differences may results in part or entirely from pre-treatment differences in tree size or site quality between designated crop and non-crop trees. Statements about observed differences between crop and non-crop trees three years following crop tree treatment are valid but attribution of these differences to crop tree management are not given the methods described. These comments apply to all Discussion sections.
317-318 State main differences for current results and those found in related research.
Conclusion
348-354 Per line 272-287 comments, the study methods do not support attribution of observed differences in fine roots or soil properties to the crop tree management treatment. Conclusion regarding fine root and soil differences three years after crop tree management treatment without attribution of the cause of the differences would be valid.
References
361 Change “masson” to “Masson” if name originates from a proper noun e.g. species name originated for a person named Masson.
378 and 395 Eissenstat reference repeated
397 Change to “close-to-nature silviculture”
414 Change to “and”
415 First letter should not be capitalized for article titles.
426 Change “capitalize” to “Capitalize”
430 Change “fine” to “Fine”
